# Booster Doses of Anti COVID-19 Vaccines: An Overview of Implementation Policies among OECD and EU Countries

**DOI:** 10.3390/ijerph19127233

**Published:** 2022-06-13

**Authors:** Fabrizio Bert, Giacomo Scaioli, Lorenzo Vola, Davide Accortanzo, Giuseppina Lo Moro, Roberta Siliquini

**Affiliations:** 1Department of Public Health Sciences and Pediatrics, University of Turin, 10126 Turin, Italy; fabrizio.bert@unito.it (F.B.); giacomo.scaioli@unito.it (G.S.); lorenzo.vola@unito.it (L.V.); davide.accortanzo@unito.it (D.A.); roberta.siliquini@unito.it (R.S.); 2AOU City of Health and Science of Turin, 10126 Turin, Italy

**Keywords:** health care policy, booster dose, COVID-19, vaccine

## Abstract

The need for an anti-COVID-19 booster dose posed an organizational challenge for health policy makers worldwide. Therefore, this study aimed to explore the health policies regarding the booster dose through an overview of recommendations issued in high-income countries. Between 10 November and 16 December 2021, the authors searched for state-level official documents about the offer of the booster dose, considering the 43 countries belonging to the European Union (EU) or the Organisation for Economic Co-operation and Development (OECD). Mainly due to the lack of English translation, 15 countries were excluded. A total of 135 documents were selected. Almost all the countries started administering the booster dose between September and November 2021. The most used products were mRNA vaccines, followed by Vaxzevria-AstraZeneca and Jcovden-Janssen/Johnson & Johnson. All countries established criteria to define categories of individuals to be vaccinated as a priority. A six/five-months interval was the main choice for general population vaccinated with mRNA vaccines, while shorter intervals were chosen for vulnerable individuals or other vaccines. Despite diversities related to the differences in health systems, economical resources, and population numbers, and the need to adapt all these factors to a massive vaccination campaign, a progressive convergence towards the same vaccination policies was highlighted.

## 1. Introduction

The SARS-CoV-2 pandemic has forced the world population to face some important challenges. One of these is certainly the vaccination campaign against COVID-19, one of the most impressive vaccination campaigns in history [1]. As of 31 December 2021, 49% of the world population had full primary series against COVID-19 (i.e., received a single-dose vaccine or both doses of a two-dose vaccine) and 9% had been partially vaccinated (i.e., received only one dose of a two-dose vaccine), with over 9 billion doses administered globally [2].

Up to date all approved COVID-19 vaccines have proved to have great efficacy against the original strain and the main variants of concern (VOC) [3,4,5]. The majority of the largest studies published on this topic were about mRNA vaccines, which reported the greatest efficacy (>90%) in the prevention of symptomatic cases [3]. The mRNA vaccines and Vaxzevria-AstraZeneca showed the highest effectiveness both against symptomatic infection (>75% against Alpha/Beta/Gamma variants and >47% against Delta) and against hospitalizations and deaths (>80% against Alpha/Beta/Gamma/Delta variants) [3].

However, long-term effectiveness of COVID-19 vaccines is still not known, and data suggest waning immunity after COVID-19 vaccinations. Indeed, a drop over time in antibody titers has been reported [6,7] and it has been shown that reduced antibody titers may be associated with lower protection [8,9,10]. Moreover, studies, focusing mostly on the SARS-CoV-2 Delta variant, reported an association between time-from-vaccine and incidence of breakthrough infections (i.e., affecting fully vaccinated people) and severe COVID-19 [11,12,13]. Also, modelling of predicted efficacy against SARS-CoV-2 variants over time found that symptomatic infection protection may be reduced below 50% in the first year after some vaccines [8]. A recent systematic review and meta-regression analysis found that vaccine efficacy or effectiveness against severe disease decreased on average 8–10 percentage points between one and six months after full vaccination, but it remained above 70%. Instead, efficacy or effectiveness against infection or symptomatic disease was reduced by about 20–30 percentage points during six months from vaccination [14].

Given the loss of effectiveness of vaccines against COVID-19 over time, in the second half of 2021 scientists, public health institutions and governments started to discuss the administration to the population of a booster dose of vaccine, i.e., a beyond dose prescribed by the original vaccine protocol [8,15]. Data on safety and immunogenicity of a booster dose have been confirmed for most vaccines against COVID-19, showing boosting antibody and neutralizing responses [16,17,18,19]. Regarding effectiveness, several studies from Israel revealed that booster doses can be effective in protecting against infection, severe COVID-19, hospitalization and COVID-19-related death [20,21,22].

Worldwide, as of 31 December 2021 over 500 million booster doses have been administered (mostly in high-income and upper middle-income countries) [2,23]. Booster doses represented around 1% of global daily COVID-19 administered doses during the months from July to September 2021, then gradually increasing up to 8% of daily doses during the first days of November 2021. Between 4 and 5 November 2021, booster doses greatly increased from 8% to 17% and, since then, booster doses have been steadily above 15% of daily administered doses, thus highlighting a change in vaccination campaigns [2]. Nevertheless, a public discussion on the booster dose has been carried out both considering the still uncertain existing evidence, e.g., considering the best time when the booster dose should be administered to get the highest advantage, and the global implications on vaccine equity regarding the appropriateness (both scientific and ethical) of offering booster doses when a part of world was struggling to complete the first vaccination cycle [15,24]. 

In this context, the World Health Organization (WHO) in December 2021 stated that the introduction of booster doses should be evidence-driven and targeted to subgroups at highest risk of severe consequences and necessary to protect the health system [23]. These scientific and ethical considerations, in addition to the need of adequate stocks, national supply and implementation logistic, might lead to divergent implementation strategies of booster programmes between countries, potentially resulting in different health outcomes [24].

Thus, the main aim of the present paper was to describe and compare booster doses programmes in high-income countries (i.e., countries that were the first to provide booster doses) through an overview of official documentation released in countries belonging to the European Union (EU) and/or the Organisation for Economic Co-operation and Development (OECD). In the light of potential future situations that may occur, the final goal of this work was to be a starting point for assisting stakeholders and health policy makers, and for supporting future research that will assess the effectiveness of booster doses campaigns.

## 2. Materials and Methods

### 2.1. Data Sources and Search Strategy

To provide an overview of the strategies implemented to offer the booster dose against COVID-19, it was decided to consider the definitions and terminology used by the WHO, i.e., “Booster doses are administered to a vaccinated population that has completed a primary vaccination series (currently one or two doses of COVID-19 vaccine depending on the product)”. Therefore, the so-called “additional doses”, which are administered to extend the primary vaccination series in specific sub-groups that may have an insufficient immune response, were excluded [23].

The authors decided to take into account the strategies adopted by countries that are members of the EU (27 countries) or of the OECD (38 countries), which share common standards and practices and may have comparable organizational systems [25,26]. 

Thus, the authors searched for information about the booster dose among the 43 countries belonging to the EU or the OECD (overlap of 22 countries between the two organizations). For each country, the authors primarily searched the websites of the government, Ministry of Health and national health institutes for state-level official documents about the offer of the booster dose. Then, if no information was found, the websites of governmental webpages specific for COVID-19 or COVID-19 vaccination information, national public health societies or other national medical associations were searched. 

The search was carried out between 10 November and 16 December 2021. No restriction on the publication date of the documents was applied. 

### 2.2. Document Selection and Data Extraction

The authors considered eligible documents reporting official information about the anti-COVID-19 booster dose vaccination campaign. Specifically, documents were included if they reported information about at least one of the following features: date of beginning of the booster dose campaign; criteria of eligibility for receiving the booster dose; vaccine used for the booster dose; ways of reservation of the booster dose; places of administration of the booster dose; national/regional/local organization; restrictive measures implemented and compulsory vaccination policies.

Based on the authors’ skills, only documents written in English, French, Spanish, Italian or Portuguese were included. Documents reporting only information about the “additional dose” were excluded. 

Two authors (LV and DA) independently screened the above-mentioned websites to identify relevant documents according to the inclusion and exclusion criteria. Disagreements were solved by a consensus-based discussion with the other authors. 

After the selection of the relevant documents, the authors extracted, by using an evaluation grid, the following data about the booster dose (when available): starting date; eligibility criteria, e.g., age categories, healthcare workers (HCWs), vulnerable and high risk groups (VHRGs), long term care facilities residents (LTCF-Rs); authorized vaccines; interval between doses; dosage; other relevant information. Clarifications of the definitions of the above-mentioned terms are provided in Appendix A.

Findings from the included countries were compared and presented in tables. 

## 3. Results

### 3.1. Included Countries

Out of the 43 countries considered in our study, 15 countries were excluded (Figure 1). Among those countries, either there was no website translation, or the English translated website did not contained relevant information on COVID-19 vaccine booster dose. Therefore, the remaining 28 countries were included in the study (Figure 1), with a total of 135 official documents. A detailed list of all included documents is presented in Appendix A.

### 3.2. Date of Start of the Booster Campaign

The booster campaign started with a different timing across the countries. All the 28 states included in the study started to administer the booster dose before the starting date of the present project. However, each country observed a different policy. The first one who started the booster campaign was Israel on 30 July 2021, followed by Chile on the 11 August 2021 and, shortly after, Iceland on 16 August 2021. Then, 10 states started administering the booster dose during September 2021, 10 states during October 2021, and four states during November 2021. Specific information on the booster start date of one country was not available (Table 1).

### 3.3. Product

#### 3.3.1. Product Administered

Concerning the product administered (Table 2), all the 28 countries administered the booster shot with an mRNA vaccine. Some countries, however, chose a different employment of mRNA vaccine, based on variables such as age and clinical characteristics or the vaccine used to complete primary series. Four countries (Belgium, France, Germany and Ireland) chose to administer preferably the mRNA vaccine Comirnaty-Pfizer/BioNTech to people younger than 30 years old; four countries (Australia, Chile, Costa Rica, New Zealand) did not use the mRNA vaccine Spikevax-Moderna at all; two countries (Czech Republic and Cyprus) decided to administer exclusively Comirnaty-Pfizer/BioNTech to those who completed the primary vaccination series with Comirnaty-Pfizer/BioNTech; Cyprus also administered Spikevax-Moderna to those who completed the primary vaccination series with Spikevax-Moderna; Israel used Comirnaty-Pfizer/BioNTech for people aged over 12 years and Spikevax-Moderna for over 18 years.

Aside from mRNA vaccines, other products have been used. Six countries (Australia, Canada, Chile, Costa Rica, New Zealand, UK) also administered Vaxzevria-AstraZeneca, in addition to mRNA vaccines, as booster dose: Australia agreed to use Vaxzevria-AstraZeneca for patients who had received the same vaccine for the first two doses, or in case of adverse reaction to mRNA vaccine; Chile administered Vaxzevria-AstraZeneca, in the first phase of the vaccine campaign, only to people over 55 years, switching later to administration of only Comirnaty-Pfizer/BioNTech to all patients; New Zealand used either Comirnaty-Pfizer/BioNTech or Vaxzevria-AstraZeneca; Canada approved the Vaxzevria-AstraZeneca or Jcovden-Janssen/Johnson & Johnson off-label use, only when other authorized COVID-19 vaccines are contraindicated or inaccessible; Costa Rica used Vaxzevria-AstraZeneca to vaccinate workers who showed up voluntarily; UK used Vaxzevria-AstraZeneca for those who had contraindication to mRNA vaccines and had received Vaxzevria-AstraZeneca vaccine in the primary course. 

Few information on the use of Jcovden-Janssen/Johnson & Johnson to carry out the booster shot have been acknowledged: only Cyprus and the USA clearly approved the use of Jcovden-Janssen/Johnson & Johnson for this purpose.

#### 3.3.2. Dosage

Out of the countries considered, 18 countries (Australia, Belgium, Canada, Chile, Estonia, Finland, France, Iceland, Ireland, Italy, Luxembourg, Netherland, New Zealand, Norway, Spain, Sweden, Switzerland, UK) provided explicit and direct data on the dosage administered: all of them agreed on full dose (30 μg) of Comirnaty-Pfizer/BioNTech, full dose Vaxzevria-AstraZeneca (0.5 mL), half dose for Spikevax-Moderna (50 μg). Canada specified that half dose of Spikevax-Moderna is intended for people aged < 70, while a full dose was necessary for people > 70 years old and immunosuppressed users (Table 3).

### 3.4. Eligibility Criteria

Patients’ characteristics, such as age, belonging to a specific health risk class, being a HCW or being a LTCF resident influenced the choice of the vaccine used, the age cut-off at which booster dose is recommended, and the time interval between the first immunization cycle and the booster dose.

#### 3.4.1. Age

Taking into consideration the age group, most of the countries (23) started administering the booster dose to people over 60–65 years: Czech Republic, Finland, Iceland (for those who completed primary series with a mRNA vaccine), Ireland, Israel, Republic of Korea started administering the booster dose to people over 60 years; Belgium, Costa Rica, Cyprus, Estonia, France, Malta, Portugal, Switzerland and USA over 65 years; Spain and Canada over 70 years; Luxembourg over 75 years; Italy, The Netherlands, Sweden and Chile to over 80 years; Denmark, Norway over 85 years. Six countries started offering booster vaccination to young people or adults, still prioritizing older people over the younger. Among those, the UK started administering the booster dose to people over 50 years; Germany, Australia and New Zealand over 18 years; Iceland over 16 years (to those who completed the first series with Jcovden-Janssen/Johnson & Johnson). Most of the policies gradually reduced the age cut-off ending up vaccinating broader range of age groups: Luxembourg later addressed the booster dose to people over 65 years; Spain to over 60 years; Canada to over 50 years; Chile to people over 45 years; Cyprus to people over 30 years; Italy, Belgium, Denmark, Estonia, France, The Netherlands, Sweden and Norway and USA over 18 years; Iceland and Israel set the age cut-off over 12 years. Without taking into consideration the health risk class, profession or other characteristics, 13 countries (Canada, Chile, Cyprus, Czech Republic, Finland, Ireland, Luxembourg, Malta, Portugal, Spain, Switzerland, Republic of Korea, UK) have not included the whole adult population up to the end of data searching on 16 December 2021 (Table 3). Additional information is reported in Appendix A.

#### 3.4.2. Workers

Considering the policies regarding workers, 26 countries took into consideration and prioritized the booster shot in HCWs. Four countries (Czech Republic, The Netherlands, Portugal and UK) offered the booster shot to social workers and one (Portugal) also to firefighters involved in patient transport (Table 3). Additional information is reported in Appendix A.

#### 3.4.3. Clinical Characteristics

As for patients’ clinical characteristics, 24 countries specified the need to vaccinate VHRGs as soon as possible. Four countries, Luxembourg, Malta, Portugal, Israel, either did not specify any policy or specifically did not recommend the booster dose to VHRGs (Table 3). Additional information is reported in Appendix A.

#### 3.4.4. LTCF Residents

The other group that has been considered for the booster dose consists of LTCF residents. Twenty-four countries chose to give priority to LTCF residents, four countries (Czech Republic, Malta, Israel and UK) did not specify any policy for LTCF residents (Table 3). Additional information is reported in Appendix A.

### 3.5. Time Interval

As for the time interval between the primary vaccination series and the booster dose, different choices have been made mainly based on the vaccine used to complete the first cycle and the characteristics of the patient. A total of 16 countries (Australia, Canada, Cyprus, Czech Republic, Finland, France, Germany, Ireland, Italy, The Netherlands, New Zealand, Norway, Republic of Korea, Sweden, Switzerland, UK) chose a six months interval; Chile a four months interval (later extended to six months), and Malta a three months interval regardless of the product used for primary vaccine cycle. Among the above-mentioned countries that have initially chosen the six months interval, three nations (Italy, Czech Republic, France) later reduced the minimum interval to five months. Other countries specified different intervals depending on the products used in the first cycle: if the first primary series was conducted with an mRNA vaccine, six countries (Belgium, Estonia, Israel (Spikevax-Moderna), Luxembourg, Spain, USA) established a six months interval, while three (Iceland, Israel (Comirnaty-Pfizer/BioNTech), Portugal) chose a five months interval. Three countries gave specific guidelines to those who were administered Vaxzevria-AstraZeneca at the first immunization cycle: Estonia chose a five months interval; Belgium four months; Spain three months. Seven countries gave specific guidelines to those who got administered Jcovden-Janssen/Johnson & Johnson for the primary series: Estonia indicated a minimum interval of five months; Belgium initially a four months interval, later shortened to two months; Portugal and Spain three months; Luxembourg and USA two months; Iceland four weeks. Other variables influenced the choice of the minimum interval. For instance, two countries took into account patient clinical characteristics, shortening the interval to two months for immunosuppressed patients (Chile) or to three months for patients affected by a high-risk chronic disease (France). Sweden, on the other hand, shortened the interval to five months for people over 65 years. The time interval data of two countries, Denmark and Costa Rica, was not available. Additional information is reported in Appendix A.

### 3.6. Other Information

Further information that could represent important notions to citizens about the booster dose campaign were investigated. A total of 25 countries explained how-to or who-to contact to book an appointment for a booster shot. All the 28 countries also specified the location where they could receive the booster and/or who would be entitled to the administration of it. Twenty-one countries specified the presence or absence of vaccine obligations. Fifteen countries chose to delegate subnational organizations for the implementation of the booster dose campaign, including reservations and administration; some countries instructed municipalities and other regions. A unique case is Israel, which appointed HMO (health maintenance organization, mandatory public insurance company) for the vaccine campaign management.

## 4. Discussion

This study aimed to explore the health policies regarding the booster dose against COVID-19, providing an overview of the recommendations that regulate it in high-income countries. The final goal of the present paper was to represent a potential starting point for assisting stakeholders and health policy makers during similar future situations and for giving a basis to future research aiming to evaluate the effectiveness of booster dose campaigns. Indeed, in this work, a total of 28 countries belonging to either the OECD or the EU were selected and information from state level documents or governmental websites on different parameters that outline the policies for the administration of the booster dose was gathered. 

First, it is worth noting that three countries (Israel, Chile, Iceland) anticipated all the others in the booster campaign, starting the administration of the booster dose during summer 2021, while most of the countries started in autumn of the same year. Many national authorities probably preferred to wait for evidence on the booster dose in order to justify and begin the booster campaign. That could explain the fast increment of booster initiatives in the late 2021, the latency between the firsts and the ones that followed, and the similarity of the starting indications among different countries. The first available studies about booster doses were published mainly on Israeli data, and proved the booster efficacy in reducing infections and severe COVID-19 disease [22,27]. However, although it was clear that immunity towards COVID-19 wanes over time, it was unclear if the booster shot was necessary and how to regulate the administration (product, dosage, timing) [28,29].

It should be considered that the global scenario about COVID-19 developed rapidly, leading to new evidence on diagnosis, treatment and prevention of this new disease. While, in the case of primary series vaccination against COVID-19, countries were waiting for the development and approval of COVID-19 vaccines and there was an obvious lack of evidence about the long term efficacy of these drugs, for the booster campaign the decisions have been taken more quickly by countries such as Israel, Chile and Iceland. In early July 2021, the need to administer a booster dose was taken into consideration from several international agencies, although not yet recommended, due to lack of evidence [30]. Later in August 2021 the “Joint Statement from HHS Public Health and Medical Experts on COVID-19 Booster Shots” concluded that a booster shot will be needed to maximize vaccine-induced protection and prolong its durability [31]. In early September, EMA and FDA started communicating the need for a booster dose: on the 1 September 2021 the FDA press release announced a plan to prepare for additional COVID-19 vaccine doses, or boosters [32]. On the 2 September 2021 EMA press release reported no urgent need for the administration of booster doses but that a preparatory plans for administering boosters may be considered [33]. That could have been a turning point in country policy making. The 22 September 2021 the FDA news release confirmed the amendment to the EUA to include a single booster dose to Pfizer Inc [34]. On 4 September 2021 EMA press release cited “On the basis of this data, the Committee concluded that booster doses may be considered at least 6 months after the second dose for people aged 18 years and older. At national level, public health bodies may issue official recommendations on the use of booster doses, taking into account emerging effectiveness data and the limited safety data.” [35]. Up to 16 December 2021, roughly half of the countries considered had already started the booster campaign. Later on, other products have been approved for booster dose. On 25 October 2022, the EMA human medicines committee (CHMP) concluded that a booster dose of the COVID-19 vaccine Spikevax-Moderna may be considered in people aged 18 years and above [36]. On 25 October 2022 the CHMP concluded that a booster dose of COVID-19 Vaccine Jcovden-Janssen/Johnson & Johnson may be considered at least two months after the first dose in people aged 18 years and above [37].

The almost simultaneous start of the booster dose campaign in many countries in autumn 2021 was surely promoted or facilitated by the international agencies approvals and recommendations listed above. Moreover, the COVID-19 winter surge, which was starting to show early in September, ref. [2] could have played a role in the timing of policy making.

Focusing on eligibility criteria, all the countries specified an age cut-off for the booster dose, and almost all the countries started administering the vaccines from people 60–65 years and older, prioritizing the elderly. Most of the countries gradually lowered the age cut-off, and the majority of them ended up including the entire adult population. Some countries immediately specified that the goal of the booster campaign was to vaccinate the entire adult population, but they opened vaccinations in stages, starting from elderly up to involving the entire adult population. In few cases, it has even been considered from the beginning to administer the booster dose to all the population over 12 years. Therefore, in some cases the vaccinations were opened gradually, starting from an age cut-off without specifying what the objective of the campaign was, in the other case the objective was clearly specified from the beginning. These different strategies could lead to differences from the recipient’s point of view: the population to which the information is addressed could perceive the information differently in one case or the other. Since the administration of a COVID-19 vaccine booster dose has been demonstrated to increase the immune responses and was anticipated to increase protection, especially in older people, it is reasonable the choice to prioritize the elderly, whose immunity wanes faster over time and the probability of inauspicious course of the COVID-19 disease is higher [38,39,40].

As for the workers, almost all the countries gradually included HCWs, and in most cases the booster dose was offered from the beginning of the campaign. Many are the reasons that could lead to this decision, probably it is partially because HCWs are exposed to a higher infection risk, but also in order to protect health care users, since the protection against transmission from vaccinated individuals who are infected also appears to wane over time [41,42]. Presumably, for the same reasons, some countries also decided to include social workers and, only in Portugal, also firefighters employed in patient transport. Booster doses have proven a moderate impact on reducing risk of developing infection from COVID-19; therefore ensuring HCWs have received three doses of vaccine has an impact on reducing transmission to hospitalized patients or LTCF-Rs by preventing infection in workers [43,44,45].

Considering the other worker categories, it should be considered that, since the first phase of the pandemic, many individuals started working from home, thus avoiding social interactions and minimizing the risk of transmission. However, in addition to HCWs, many are the key workers (e.g., law enforcement, those essential to the provision of food and other key goods) that have no possibility for remote working, thus being not able to comply with the rules of isolation and being exposed to higher risk of COVID-19 disease [46]. Evidence showed a higher risk of COVID-19 infection in workers during the pandemic; therefore policy makers should take into consideration that other workers, besides HCWs, should be given priority in the booster campaign or future vaccine campaigns [47,48]. 

Another main parameter to prioritize the booster dose was the patients’ clinical characteristics. Overall, patients affected by conditions that imply a greater susceptibility to COVID-19 or a greater probability of unfortunate evolution have been taken into consideration among almost all countries. Each country chose a specific and different definition of VHRGs, ending up including different patients and creating, almost in each country, a different definition of VHRG. However, generally VHRGs has been considered patients with higher risk of COVID-19 severe disease including immunosuppressed patients, people suffering from cardiovascular diseases, respiratory diseases, and diabetes. In this study only the booster dose was considered; however, it should be mentioned that there could be an overlap between the booster dose and additional dose policies, because for some vulnerable patients the third dose falls within additional dose, while for others within the booster dose policy; sometimes it is difficult, especially in clinical settings, to get whether some patients are eligible for the additional or the booster dose, each following different recommendations. It should be clarified that additional dose and booster dose follow different rationales, but from a practical point of view it could be generally said that a shorter interval was chosen between the second and third dose in VHRGs; this operative concept was well developed by the Iceland Directorate of Health, which used a simplified booster definition and publish a plan of action in one place [49]. 

Most of the countries took into consideration, for the booster dose, LTCF-Rs, which are particularly vulnerable to severe and fatal forms of COVID-19 infection [50,51]. Although the majority of LTCF-Rs are eligible for age cut-off or clinical characteristics, most of the countries chose to specifically focus on LTCF-Rs, probably because, other than age and clinical vulnerability, there might be a higher epidemiological risk of severe COVID-19, since such vulnerable patients are gathered in one cluster and are linked to higher risk of healthcare associated infections [51]. The impact of vaccination in LTCFs was still unknown in November 2020 [52], then evidence clearly showed consistent reduction of the COVID-19 burden due to booster vaccination in LTCFs [53]. 

Eligibility criteria for the COVID-19 vaccine campaign, as the ones mentioned above (age, HCWs, LTCF-Rs), do overlap in some cases. Indeed, among the countries that have not included HCWs, some countries decided to administer the booster dose to the entire adult population, thus meaning that the HCWs were included even if not prioritized. The same consideration can be made for VHRGs and LTCF-Rs. Some countries decided to release recommendations as simple and concise as possible, while others decided to emphasize the need for immunization in certain categories.

A large consensus has been highlighted regarding the product to be administered for the Booster Dose, since all countries used at least one mRNA vaccine, while only two also administered or considered the use of Jcovden-Janssen/Johnson & Johnson and six Vaxzevria-AstraZeneca (Table 2). Moreover, the choice of the product was generally stable over time, with few countries who chose to consider different products or ceased to use one. On the other hand, it has been observed an evolving, but quite homogeneous, pattern in the modality of the administration over time, observing mainly changes about the choice of time interval between the primary series and the booster dose and also changes in the eligibility criteria to receive the booster dose with one product or another. 

The issued recommendations for the booster campaign often followed international agencies approval. For instance, EMA and ECDC agreement on ‘mix-and-match’ approach for both initial course and boosters guided, in many countries, the decision to administer as booster dose any mRNA vaccine regardless of the vaccine administered in the primary series [54]. The “Spikevax: EMA recommendation on booster” has been welcomed by several states outlining the dosage for the Spikevax-Moderna vaccine [36]. In addition, the “EMA evaluating data on booster dose of COVID-19 Vaccine Janssen” [55] and the “COVID-19 Vaccine Janssen: EMA recommendation on booster dose” [37] outlined the use of Jcovden-Janssen/Johnson & Johnson as booster.

The time interval between the primary series and the booster dose depended mainly on the product used to complete the primary series; most countries also adjusted the interval over time. For people who completed the primary series with an mRNA vaccine, most countries chose a six months interval, later reduced in some cases to five months interval. For those who received Vaxzevria-AstraZeneca or Jcovden-Janssen/Johnson & Johnson, the interval was generally shorter, especially Jcovden-Janssen/Johnson & Johnson with intervals mainly between two and three months. Even though not all countries considered VHRG patients for determining the time interval, among countries that took them into consideration, it was agreed on a shorter interval to guarantee a better protection of fragile patients. 

Last, it is worth noting that the global scenario is still evolving fast, indeed, since its first appearance in November 2021, the Omicron variant has rapidly spread even among vaccinated populations leading to concerns about the effectiveness of current vaccines [56]. The emergence of new variants of SARS-CoV-2 points out new challenges in the development of new vaccines against SARS-CoV-2 and in the identification of new strategies for the control of the transmission [57]. Evidence has shown effectiveness of current vaccines on containing the pandemic also considering the Omicron variant, but such effectiveness seems dose-dependent and the immunity wanes over time; therefore, a booster dose can increase protection against COVID-19 [58,59]. An assumption made by Khan NA et al., interpreting research conducted by Pfizer [9], estimates an increase in protection by 10% due to the booster dose [56]. Further studies have shown that COVID-19 booster doses do provide protection against Omicron variant [60], but questions on if and how the vaccination campaign should be developed are still ongoing [61]. Moreover, since new and different vaccines are currently available, doubts remain whether homologous or heterologous vaccination and, in the case of the heterologous choice, which combination of vaccines may be more effective in preventing COVID-19 [3]. In addition, literature suggests that, because of the waning of immunity and the emergence of new variants, new vaccines, including variant-specific ones, should indeed be developed [58]. Thus, innovative and specific studies are needed to provide evidence on currently used and new vaccines and possible strategies to be implemented to tackle new variants. 

This study, providing an overview of health policies regarding the booster dose, could be useful to policy makers and health professionals in making further recommendations. Different topics have been addressed. Moreover, in light of future data on the effectiveness of the booster dose throughout the considered countries, this study could be a starting point to evaluate the effectiveness of different booster vaccination campaigns, comparing similar approaches in order to highlight which recommendations were most effective in preventing the COVID-19 burden. Indeed, evidence about best practice for booster dose campaigns is still currently poor, even if some data shed light on certain characteristics that a booster strategy should have [62,63,64]. Specifically, it has been reported that the ‘mix-and-match’ approach seems to be more effective and a realistic policy to be implemented for booster campaigns [62,63,64]. However, the best strategies and dosing intervals to provide maximum benefit, especially in the context of the new variants spread, still need to be clearly defined [63]. Similarly, although many policies define priority groups to administer vaccines as reported in our results, a systematic review concluded that the current evidence on prioritization is still insufficient and there is need of adequately powered research assessing this issue [65,66]. 

Furthermore, beyond best strategies for administration, we would like to argue about best practice in communication. A comprehensive paper published by Ratzan and colleagues highlighted that planning ahead (while also acknowledging the unpredictability of these circumstances) and the focus on people are essential features for an enhanced health communication [67]. In light of these relevant considerations, our results suggest two main reflections. First, the planning ahead strategy was not implemented in several countries: even if some countries did declare at the beginning of the campaign the steps of the vaccination priorities, other countries did not make public the entire strategy of the campaign early. As planning ahead is fundamental for a transparent health communication, the lack of it may worsen the already reduced trust in governments that the general population is experiencing during the pandemic [68,69]. In addition, the focus on people seems to be limited considering that many countries were excluded from the paper due to the absence of detailed information in English. Indeed, many individuals (such as visitors, international students, foreign workers, and so on) result to be kept out from relevant information for their health.

The present work has some strengths and limitations that should be acknowledged. Indeed, this study took into consideration many countries, giving a realistic assessment of health policies regarding the booster dose in high-income countries. Despite the broadness of the topic, the search has been rapidly conducted thus making it possible to describe a situation that is still current today such as the need for evaluation of a fourth dose of COVID-19 vaccine. Above all, language represented one of the main limitations, restricting the countries included in the study and the sources of information used. The researchers highlighted the difficulty to retrieve univocal and certain state-level information. In addition, considering only state-level guidelines, further details and differences that could potentially be highlighted in regional or local guidelines were missing. During the data retrieval the researchers noticed a discrepancy between state-level documentation and other sources (mostly online journals and news). Therefore, the choice of considering only state level documentation, although it can be considered good quality information, might not report the real entry into force of the recommendations. For instance, it is difficult to be certain about the date each booster campaign started, given the gap between the ministerial indications and the actual implementation of such indications. It shall also be acknowledged that during the course of the search the vaccine campaigns rapidly evolved, therefore the data collected for each country are not all referred to the same day. 

## 5. Conclusions

In conclusion, a heterogeneous pattern of health policies has been found in the selected countries. Those differences might be necessary to adapt to different realities and comply with different health systems, economical resources, and the unpredictable vaccines availability. This led to a quite heterogeneous pattern of policies in the global scenario. However, despite many differences, vaccine campaigns over the studied countries moved in the same direction towards the same goal. 

## Figures and Tables

**Figure 1 ijerph-19-07233-f001:**
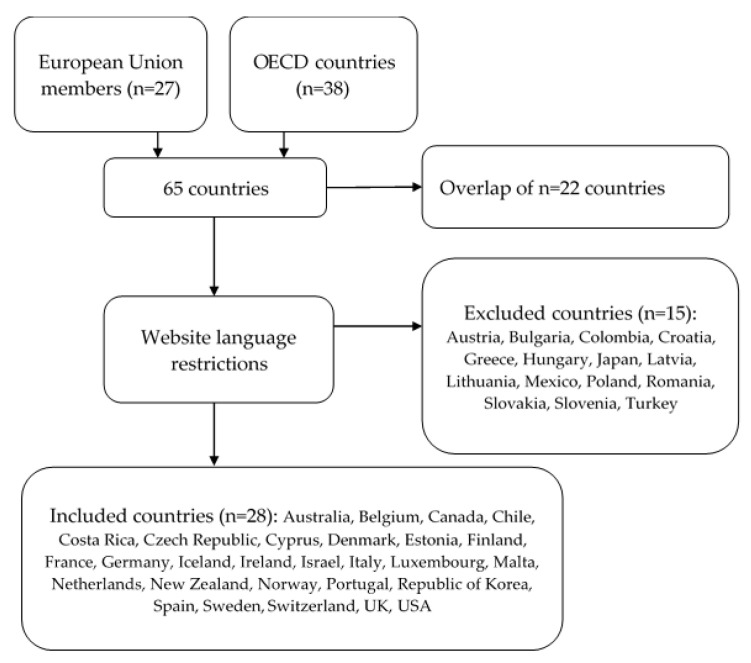
Countries included in the study.

**Table 1 ijerph-19-07233-t001:** Starting date of booster dose campaign.

Country	Date
Israel	30 July 2021
Chile	11 August 2021
Iceland	16 August 2021
Finland	September *
Cyprus	September *
France	1 September 2021
Luxembourg	14 September 2021
Czech Republic	20 September 2021
Belgium	24 September 2021
United States of America	24 September 2021
Italy	27 September 2021
Denmark	29 September 2021
United Kingdom	30 September 2021
Ireland	October *
Malta	October *
Portugal	October *
Republic of Korea	October *
Sweden	2 October 2021
Norway	5 October 2021
Estonia	12 October 2021
Spain	25 October 2021
Costa Rica	26 October 2021
Canada	29 October 2021
Australia	8 November 2021
Switzerland	15 November 2021
Netherlands	19 November 2021
New Zealand	29 November 2021
Germany	Information not retrieved

* Aspecific information for the date of start of the campaign.

**Table 2 ijerph-19-07233-t002:** Product used for the booster dose campaign against COVID-19 (countries might have used more than one product).

Product	Countries
Comirnaty-Pfizer/BioNTech	*n* = 28 (Australia, Belgium, Canada, Chile, Costa Rica, Czech Republic, Cyprus, Denmark, Estonia, Finland, France, Germany, Iceland, Ireland, Israel, Italy, Luxembourg, Malta, The Netherlands, New Zealand, Norway, Portugal, Republic of Korea, Spain, Sweden, Switzerland, USA, UK)
Spikevax-Moderna	*n* = 24 (Belgium, Canada, Czech Republic, Cyprus, Denmark, Estonia, Finland, France, Germany, Iceland, Ireland, Israel, Italy, Luxembourg, Malta, The Netherlands, Norway, Portugal, Republic of Korea, Spain, Sweden, Switzerland, USA, UK)
Vaxzevria-AstraZeneca	*n* = 6 (Australia, Canada [off label], Chile, Costa Rica, New Zealand, UK)
Jcovden-Janssen/Johnson & Johnson	*n* = 2 (Cyprus, USA)

**Table 3 ijerph-19-07233-t003:** Administration of a booster dose of a vaccine against COVID-19: first recommendations regarding eligibility criteria and interval by country.

Country	Eligibility Criteria				Interval
	General Population	HCWs	LTCF-Rs	VHRGs	
Australia	>18 yo	yes	yes	yes	>6 months
Belgium	>65 yo	yes	yes	yes	>6 months if previous mRNA, >4 months if previous Vaxzevria-AstraZeneca, >2 months if previous Jcovden-Janssen/Johnson & Johnson
Canada	>80 yo	yes	yes	yes	>6 months
Chile	>80 yo	yes	yes	yes	>4 months >2 months if VHRG
Costa Rica	n.d.	yes	n.d.	n.d.	n.d.
Czech Republic	>60 yo	yes	yes	yes	>6 months, >5 months if VHRG
Cyprus	n.d.	n.d.	yes	yes	>6 months
Denmark	>85 yo	n.d.	yes	yes	>6 months
Estonia	n.d.	n.d.	n.d.	n.d.	n.d.
Finland	>60 yo	yes	yes	yes	>6 months
France	>65 yo	yes	yes	yes	>6 months, 3–6 months for VHRG
Germany	n.d.	n.d.	n.d.	n.d.	n.d.
Iceland	>16 yo	n.d.	n.d.	n.d.	>4 weeks for those previously vaccinated with Jcovden-Janssen/Johnson & Johnson
Ireland	>80 yo	n.d.	yes (>65 yo)	n.d.	>6 months
Israel	>60 yo	n.d.	n.d.	n.d.	>5 months (Comirnaty-Pfizer/BioNTech), >6 months (Spikevax-Moderna)
Italy	>80 yo	yes	yes	yes (>18 yo)	>6 months
Luxembourg	>75 yo	n.d.	yes	yes (on dyalisis)	>6 months (Comirnaty- Pfizer/BioNTech), >2 months (Jcovden-Janssen/Johnson & Johnson)
Malta	>65 yo	n.d.	n.d.	n.d.	>3 months
Netherlands	>80 yo	yes	yes	yes	>6 months
New Zealand	>18 yo	yes	yes	yes	>6 months
Norway	>85 yo	no	yes	yes	>6 months
Portugal	>65 yo	no	yes	no	>5 months
Republic of Korea	no	yes	no	no	>6 months
Spain	>70 yo	no	yes	yes	>6 months
Sweden	>80 yo	yes	yes	yes	>6 months
Switzerland	>65 yo	no	yes	yes	>6 months
United Kingdom	>50 yo	yes	yes	yes	>6 months
United States of America	>65 yo	yes	yes	yes	>6 months (Comirnaty-Pfizer/BioNTech), >2 months (Jcovden-Janssen/Johnson & Johnson)

Abbreviations: LTCF-Rs long term care facilities residents; HCWs healthcare workers; n.d. no data; VHRGs vulnerable and high risk groups; yo years old.

## Data Availability

All relevant data are within the paper. The documents considered in the present study are available from the corresponding author on reasonable request.

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
