# Peer review of "Booster Doses of Anti COVID-19 Vaccines: An Overview of Implementation Policies among OECD and EU Countries"

_ijerph, 2022, doi:10.3390/ijerph19127233_

Round 1

Reviewer 1 Report

The article is very interesting and focuses on a current issue of great interest for Public Health and health systems.

The Authors explore in their paper the health policies regarding the anti-COVID-19 booster dose through an overview of recommendations issued in high-income countries.

The article is well written, it is complete and the methodology is valid.

Despite diversities related to the differences in health systems, economical resources, and population numbers, and the need to adapt all these factors to a massive vaccination campaign, a progressive convergence towards the same vaccination policies was highlighted.

This study could support to policy makers and health professionals in making further recommendations for the control of pandemic.

I would suggest to standardize the references to the various anti-covid vaccines in the text and in table 2, by inserting the commercial name and manufacturer. For example, in table 2 the name of the Janssen vaccine is not indicated while for the other vaccines the trade name has been indicated but not the manufacturer.

Furthermore, I would suggest including a comment on the current pandemic scenario related to the omicron variant and its sub-variants in the discussion. Indeed, the rapid increase in Covid-19 cases due to the omicron variant also in highly vaccinated populations has aroused concerns about the effectiveness of current vaccines. At this point of the pandemic, whit the omicron strain that is supercharging the booster debate, global policy must consider the hazards of adopting booster dosages across the world (Khan NA, Al-Thani H, El-Menyar A. The emergence of new SARS-CoV-2 variant (Omicron) and increasing calls for COVID-19 vaccine boosters-The debate continues. Travel Med Infect Dis. 2022 Jan-Feb;45:102246. doi: 10.1016/j.tmaid.2021.102246. Epub 2021 Dec 21. PMID: 34942376; PMCID: PMC8687713.)

Furthermore, the emergence of new variants of SARS-CoV-2 highlights a new challenge in the development of new vaccines against the virus and in the identification of new strategies for control of the COVID-19 pandemic (Hoffmann M, Krüger N, Schulz S, Cossmann A, Rocha C, Kempf A, Nehlmeier I, Graichen L, Moldenhauer AS, Winkler MS, Lier M, Dopfer-Jablonka A, Jäck HM, Behrens GMN, Pöhlmann S. The Omicron variant is highly resistant against antibody-mediated neutralization: Implications for control of the COVID-19 pandemic. Cell. 2022 Feb 3;185(3):447-456.e11. doi: 10.1016/j.cell.2021.12.032. Epub 2021 Dec 24. PMID: 35026151; PMCID: PMC8702401).

Reviewer 2 Report

First and foremost, I wish to congratulate the authors for shedding light on such a relevant and important topic. The study aims to examine booster shots policy in countries that are either located in Europe or a part of the OECD system. Overall, I think this study bears meaningful influences on research, practice, and policymaking. Please find my comments below and respond to them sufficiently, as I believe answering these concerns could help the authors further enhance their work, and in turn, the readers better appreciate the study.

A key concern I have about the research centres on its overall relevance, largely due to the evolving nature of COVID-19 and the shifting landscape of COVID-19 vaccination.

Based on the findings of the study and the broad literature, could the authors please shed light on the practical and transferrable insights that government officials and health experts, including those in high-, middle, and low-income countries, could adopt to best promote booster shots for future COVID-19 surges or other infectious disease outbreaks? For instance, what are the dos and don’ts in terms of organizing booster shots programs? What systems should be in place to ensure the success of these programs?

A related note is that little is said about COVID-19 communication or misinformation in the study. Could the authors please shed light on what they found from their analysis in terms of the best communication practices? One would argue that, for instance, the mere fact that “Mainly due to the lack of English translation, 15 countries were excluded” might reveal deeper issues—how might visitors, international students, or foreign business people receive COVID-19 information and then take their booster shots if official information is not even available in English—arguably the most common language spoken across the world.

Reviewer 3 Report

This is a study of the application of vaccination health policy in various countries, whose main limitation is the language bias (which eliminates 15 countries) and which is explained by the authors in the limitations section. The introduction is adequate and the material and methods reproducible.

Author Response

Response to Reviewer 3 Comments

Point 1: This is a study of the application of vaccination health policy in various countries, whose main limitation is the language bias (which eliminates 15 countries) and which is explained by the authors in the limitations section. The introduction is adequate and the material and methods reproducible

Response 1: We would like to thank the reviewer for these comments.